# GPT-RE: In-context Learning for Relation Extraction using Large Language Models

**Zhen Wan** [1]    **Fei Cheng** [1]    **Zhuoyuan Mao** [1]
**Qianying Liu**[1]    **Haiyue Song**[1]    **Jiwei Li**[2]    **Sadao Kurohashi**[1]
[1] Kyoto University, Japan
[2] Zhejiang University, China
{zhenwan, zhuoyuanmao, ying, song}@nlp.ist.i.kyoto-u.ac.jp
{feicheng, kuro}@i.kyoto-u.ac.jp
{jiwei_li}@zju.edu.cn

## Abstract

In spite of the potential for ground-breaking achievements offered by large language models (LLMs) (e.g., GPT-3) via in-context learning (ICL), they still lag significantly behind fully-supervised baselines (e.g., fine-tuned BERT) in relation extraction (RE). This is due to the two major shortcomings of ICL for RE: (1) low relevance regarding entity and relation in existing sentence-level demonstration retrieval approaches for ICL; and (2) the lack of explaining input-label mappings of demonstrations leading to poor ICL effectiveness.

In this paper, we propose GPT-RE to successfully address the aforementioned issues by (1) incorporating task-aware representations in demonstration retrieval; and (2) enriching the demonstrations with gold label-induced reasoning logic. We evaluate GPT-RE on four widely-used RE datasets and observe that GPT-RE achieves improvements over not only existing GPT-3 baselines, but also fully-supervised baselines as in Figure 1. Specifically, GPT-RE achieves SOTA performances on the Semeval and SciERC datasets, and competitive performances on the TACRED and ACE05 datasets.

Additionally, a critical issue of LLMs revealed by previous work, the strong inclination to wrongly classify NULL examples into other pre-defined labels, is substantially alleviated by our method. We show an empirical analysis.[1]

## 1 Introduction

The emergence of large language models (LLMs) such as GPT-3 (Brown et al., 2020; Thoppilan et al., 2022; Chowdhery et al., 2022; Rae et al., 2021; Hoffmann et al., 2022) represents a significant advancement in natural language processing (NLP). Instead of following a pretraining-and-finetuning pipeline (Devlin et al., 2019; Beltagy et al., 2019; Raffel et al., 2019; Lan et al., 2019; Zhuang et al., 2021), which finetunes a pre-trained model on

---

[1]Codes will be released after the anonymous period.

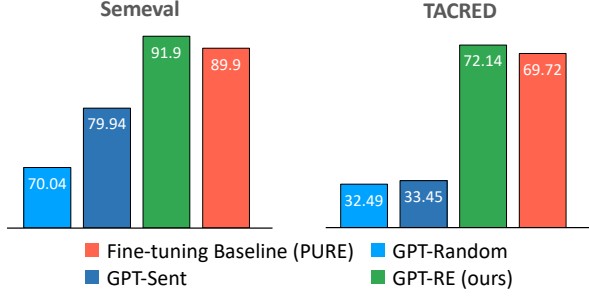

Figure 1: **Micro F1 performances on two RE datasets**. Previous GPT baselines (*GPT-Random*: randomly selected demonstrations and *GPT-Sent*: sentence-level demonstration retrieval) largely underperform fine-tuning baseline PURE while our *GPT-RE* substantially outperforms all baselines.

a task-specific dataset in a fully-supervised manner, LLMs employ a new paradigm known as in-context learning (ICL) (Brown et al., 2020; Min et al., 2022a) which formulates an NLP task under the paradigm of language generation and makes predictions by learning from a few demonstrations. Under the framework of ICL, LLMs achieve remarkable performance rivaling previous fully-supervised methods even with only a limited number of demonstrations provided in various tasks such as solving math problems, commonsense reasoning, text classification, fact retrieval, natural language inference, and semantic parsing (Brown et al., 2020; Min et al., 2022b; Zhao et al., 2021; Liu et al., 2022b; Shin et al., 2021).

Despite the overall promising performance of LLMs, the utilization of ICL for relation extraction (RE) is still suboptimal. RE is the central task for knowledge retrieval requiring a deep understanding of natural language, which seeks to identify a predefined relation between a specific entity pair mentioned in the input sentence or NULL if no relation is found. Given a test input, ICL for RE prompts the input of LLMs with the task instruction, a few demonstrations retrieved from the training data,

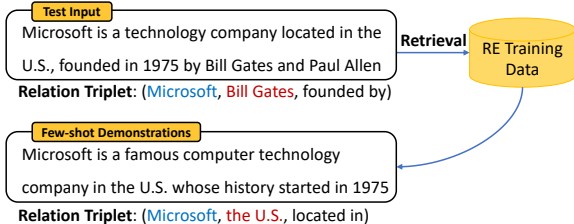

Figure 2: Retrieval without considering the task-aware triplet results in noisy demonstrations.

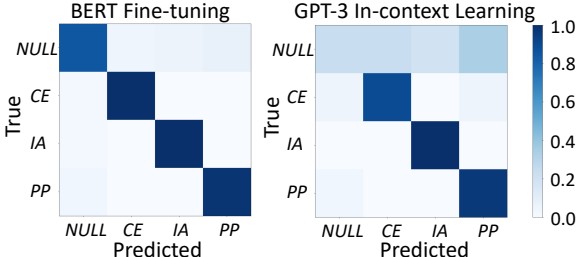

Figure 3: Confusion matrix on Semeval dataset with three selected relation labels. The NULL examples are overpredicted to other relations by GPT-3. CE: Cause-Effect, IA: Instrument-Agency, PP: Product-Producer.

and the test input itself. Then LLMs generate the corresponding relation. Recent research (Gutiérrez et al., 2022) has sought to apply GPT-3 ICL to biomedical RE, but the results are relatively negative and suggest that GPT-3 ICL still significantly underperforms fine-tuned models.

The reasons that cause the pitfall of GPT-3 ICL in RE are two folds: (1) The low relevance regarding entity and relation in the retrieved demonstrations for ICL. Demonstrations are selected randomly or via $k$-nearest neighbor ($k$NN) search based on sentence embedding (Liu et al., 2022b; Gutiérrez et al., 2022). Regrettably, $k$NN-retrieval based on sentence embedding is more concerned with the relevance of the overall sentence semantics and not as much with the specific entities and relations it contains, which leads to low-quality demonstrations. As shown in Figure 2, the test input retrieves a semantically similar sentence but is not desired in terms of entities and relations.

(2) The lack of explaining input-label mappings in demonstrations leads to poor ICL effectiveness: A vanilla form of ICL lists all demonstrations as input-label pairs without any explanations. This may mislead LLMs to learn shallow clues from surface words, while a relation can be presented in diverse forms due to language complexity. Especially when ICL has a maximal input length, optimizing the learning efficiency of each single demonstration becomes extremely important.

To this end, we propose GPT-RE for the RE task. GPT-RE employs two strategies to resolve the issues above: (1) **task-aware retrieval** and (2) **gold label-induced reasoning**. For (1) task-aware retrieval, its core is to use representations that deliberately encode and emphasize entity and relation information rather than sentence embedding for $k$NN search. We achieve this by two different retrieval approaches: (a) entity-prompted sentence embedding; (b) fine-tuned relation representation, which naturally places emphasis on entities and

relations. Both methods contain more RE-specific information than sentence semantics, thus effectively addressing the problem of low relevance.

For (2) gold label-induced reasoning, we propose to inject the reasoning logic into the demonstration to provide more evidence to align an input and the label, a strategy akin to the Chain-of-Thought (CoT) research (Wei et al., 2022; Wang et al., 2022b; Kojima et al., 2022). But different from previous work, we allow LLMs to elicit the reasoning process to explain not only why a given sentence should be classified under a particular label but also why a NULL example should not be assigned to any of the pre-defined categories. This process significantly improves the ability of LLMs to align the relations with diverse expression forms.

Recent work reveals another crucial problem named "overpredicting" as shown in Figure 3: we observe that LLMs have the strong inclination to wrongly classify NULL examples into other pre-defined labels . A similar phenomenon has also been observed in other tasks such as NER (Gutiérrez et al., 2022; Blevins et al., 2022). In this paper, we show that this issue can be alleviated if the representations for retrieval can be supervised with the whole set of NULL in the training data.

We evaluate our proposed method on three popular general domain RE datasets: Semeval 2010 task 8, TACRED and ACE05, and one scientific domain dataset SciERC. We observe that GPT-RE achieves improvements over not only existing GPT-3 baselines, but also fully-supervised baselines. Specifically, GPT-RE achieves SOTA performances on the Semeval and SciERC datasets, and competitive performances on the TACRED and ACE05 datasets.

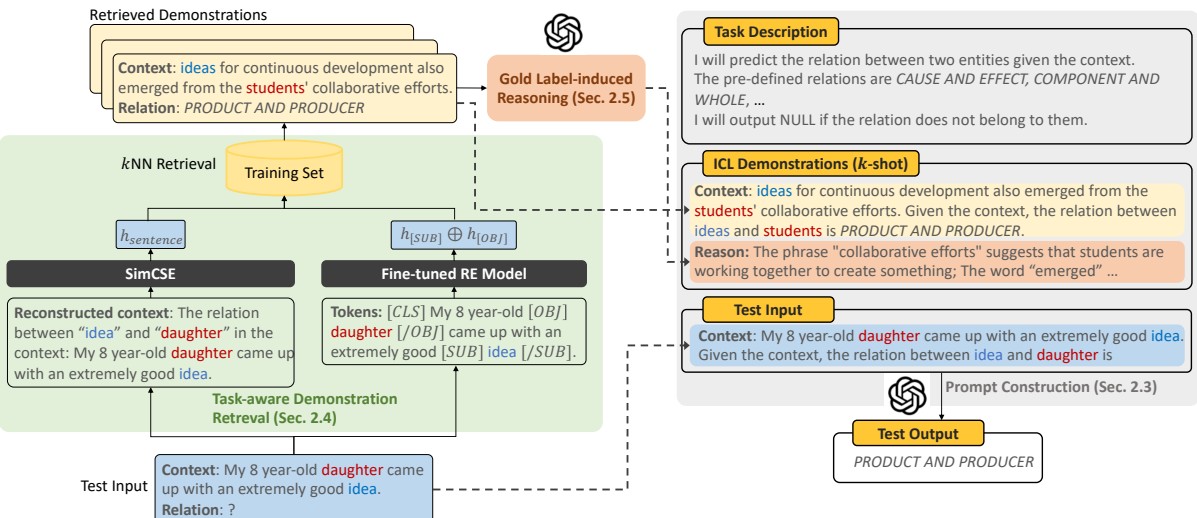

Figure 4: **An illustration of GPT-RE**. Given a test input, we first leverage two different task-aware retrieval methods to search for highly relevant demonstrations from the training set, and then incorporate the gold label-induced reasoning for each demonstration. Above contents will then be included in the prompt construction to make the prediction.

## 2  Methodology: GPT-RE

### 2.1  Task Definition

Let $\mathcal{C}$ denote the input context and $e_{\text{sub}} \in \mathcal{C}$, $e_{\text{obj}} \in \mathcal{C}$ denote the pair of subject and object entity. Given a set of pre-defined relation classes $\mathbb{R}$, relation extraction aims to predict the relation $y \in \mathbb{R}$ between the pair of entities $(e_{\text{sub}}, e_{\text{obj}})$ within the context $\mathcal{C}$, or if there is no pre-defined relation between them, predict $y = \text{NULL}$.

### 2.2  Overview

We will first introduce the prompt construction to formalize RE as a language generation task in Sec. 2.3. Then to improve the ICL framework for RE, we will introduce two modules: (1) task-aware demonstration retrieval to select higher-quality demonstrations (Sec. 2.4); (2) gold label-induced reasoning to enrich each demonstration with explanations (Sec. 2.5). In Figure 4, we show the concrete workflow of processing a test input.

### 2.3  Prompt Construction

We construct a prompt for each given test example, which is fed to the GPT-3 model. Each prompt consists of the following components:

**Instructions** $\mathcal{I}$  We provide a succinct overview of the RE task description and the set of pre-defined classes $\mathbb{R}$. The model is explicitly asked to output the relation, which belongs to the pre-defined classes. Otherwise, the model will output NULL.

**ICL Demonstrations** $\mathcal{D}$  We first leverage a task-aware retriever to acquire a $k$-shot demonstration set, then enrich each demonstration $(x_i, y_i)$ with the gold label-induced reasoning $r_i$ to build a new set of $(x_i, y_i, r_i)$ as $\mathcal{D}$.

**Test Input** $x_{test}$  Similar to the demonstrations, we offer the test input $x_{test}$, and GPT-3 is expected to generate the corresponding relation $y_{test}$.

In summary, GPT-RE can be formulated as:

$$p\left(y_{test} \in \mathbb{R} \cup \{\text{NULL}\} | \mathcal{I}, \mathcal{D}, x_{test}\right) \qquad (1)$$

### 2.4  Task-aware Demonstration Retrieval

Since ICL demonstrations closer to the test sample in the embedding space result in more consistent and robust performance (Liu et al., 2022b). Recent work (Gutiérrez et al., 2022; Liu et al., 2022b) employs the $k$NN to retrieve the most similar examples in the training set as the few-shot demonstrations for each test input. As $k$NN relies on the choice of the embedding space to encode both test input and examples in the training set, they propose to obtain sentence embedding using pre-trained language models, or other improved sentence embedding.

However, using sentence embedding for $k$NN retrieval has a severe drawback: relation extraction focuses on pair-wise entities, which diverge from the semantic meaning of the entire sentence, leading to an ambiguous retrieval using sentence embedding. In this study, we propose two novel

methods to provide more robust representations for better retrieval quality: (1) a naive entity-prompted sentence embedding in Sec. 2.4.1; (2) an advanced fine-tuned relation representation in Sec. 2.4.2.

### 2.4.1 Entity-Prompted Sentence Embedding

Given the discrepancy between sentence embedding and relation extraction, the original context is insufficient for demonstration retrieval. Considering the importance of entity information in RE, we propose reconstructing the context by incorporating entity pair information. For example, given the context "*He* has a sister *Lisa*," the reconstructed context with the entity prompted will be "The relation between 'He' and 'Lisa' in the context: He has a sister Lisa." This approach preserves both the semantic meaning of the sentence and the entity pair-centered information during retrieval. In the paper, we employ the latest robust model Sim-CSE (Gao et al., 2021) for computing sentence embedding-based similarity.

### 2.4.2 Fine-tuned Relation Representation

Compared to prompt entity information into context sentences, a more straightforward solution is to extract the relation representation from a fine-tuned RE model for retrieving demonstrations.

Current BERT-based fine-tuning methods for RE (Baldini Soares et al., 2019; Zhong and Chen, 2021; Wan et al., 2022) attempts to capture both the context information and the entity information by adding extra marker tokens to highlight the subject and object entities and their types. Specifically, given an example: "*He* has a sister *Lisa*.", the input tokens are "[CLS] [SUB_PER] *He* [/SUB_PER] has a sister [OBJ_PER] *Lisa* [/OBJ_PER]. [SEP]" where "PER" is the entity type if provided. Denote the $n$-th hidden representation of the BERT encoder as $\mathbf{h}_n$. Assuming $i$ and $j$ are the indices of two beginning entity markers [SUB_PER] and [OBJ_PER], we define the relation representation as $\mathbf{Rel} = \mathbf{h}_i \oplus \mathbf{h}_j$ where $\oplus$ stands for concatenation of representations in the first dimension. Subsequently, this representation is fed into a feedforward network for predicting the relation probability $p(y \in \mathbb{R} \cup \{\text{NULL}\} \mid \mathbf{Rel})$.

The entity markers have explicitly encoded subject and object entities and the relation representation $\mathbf{Rel}$ is naturally enriched with the entity information. We believe this approach can potentially compensate for the limitations of GPT-3 in RE. While GPT-3 ICL has a constraint of limited

Figure 5: **An illustration of adding reasoning**.

| Dataset | # Relation | # Train | # Dev | # Test (# Subset) | NULL (%) |
|---------|-----------|---------|-------|-------------------|----------|
| Semeval | 9 | 6,507 | 1,493 | 2,717 (2,717) | 17.40% |
| TACRED | 41 | 68,124 | 22,631 | 15,509 (1,600) | 79.40% |
| SciERC | 7 | 16,872 | 2,033 | 4,088 (4,088) | 90.16% |
| ACE05 | 6 | 121,368 | 27,597 | 24,420 (2,442) | 95.60% |

Table 1: **Statistics of datasets**.

demonstrations, the fine-tuning process is unbundled and can be done on the whole train data. It has two subsequent merits. First, the relation representations are directly fine-tuned to fit the RE task, which could significantly boost the overall retrieval quality. Second, the overpredicting NULL issue will be substantially alleviated because the similar NULL demonstrated can be accurately recognized by the fine-tuned model.

### 2.5 Gold Label-induced Reasoning

Recent CoT work has reported significant progress in the commonsense and numerical reasoning tasks by automatically eliciting the reasoning steps for solving a question. While in the RE task, two entities can possibly hold multiple relations, e.g., "Joe Biden" can be either the president of or lives in "U.S.". The reasoning generation could be out of focus if it lacks interaction with the gold label.

In this section, we propose to let GPT-3 induce the reasoning logic for each demonstration by the corresponding gold relation label. As shown in Figure 5, given a selected demonstration, we first generate a query prompt "What are the clues that lead to the relation between [entity1] and [entity2] to be [relation] in the sentence [context]?" based on the demonstration and subsequently ask GPT-3 to generate clues "It is because: ..." on the labeled relation between the pair of entities in the context. Finally, we augment the demonstration by incorporating the generated clues induced by GPT-3.

| Methods | Retriever | Semeval | TACRED | SciERC | ACE05 |
|---|---|---|---|---|---|
| *GPT-3 Baselines (Best k-shot)* | | | | | |
| GPT-Random | - | 70.04 (30) | 32.49 (15) | 17.92 (25) | 9.04 (25) |
| GPT-Sent | SimCSE | 79.94 (30) | 33.45 (15) | 20.96 (25) | 6.31 (25) |
| *Ours (Best k-shot)* | | | | | |
| GPT-RE_SimCSE | SimCSE | 81.02 (30) | 37.44 (15) | 26.46 (25) | 8.67 (25) |
| GPT-RE_SimCSE* | SimCSE | 77.49 (15) | 31.58 (10) | - | - |
| + Reasoning | SimCSE | 79.88 (15) | 33.18 (10) | - | - |
| GPT-RE_FT | PURE | **91.90** (25) | 72.14 (15) | **69.00** (30) | 68.73 (25) |
| GPT-RE_FT* | PURE | 91.11 (15) | 70.38 (10) | - | - |
| + Reasoning | PURE | 91.82 (15) | 70.97 (10) | - | - |
| *Fine-tuned RE Baselines* | | | | | |
| Cohen et al. (2020) | | **91.90** | - | - | - |
| Wang et al. (2022a) | | - | ♣76.80 | - | - |
| PURE (Zhong and Chen, 2021) | | 89.90 | 69.72 | 68.45 | **70.09** |

Table 2: **Main Results on four RE datasets**. All results are given by Micro-F1. * denotes the same $k$-shot for the comparison with + Reasoning. Due to the costly GPT-3 expense, we conducted Reasoning experiments on the two relatively smaller datasets Semeval and TACRED. ♣ denotes that this performance is not comparable as it evaluates on the entire test set. The underline denotes the results outperforming the fine-tuning baseline PURE.

## 3 Experiment Setup

### 3.1 Datasets

We evaluate on three popular general domain RE datasets and one scientific domain dataset. Due to the cost of running the model in the API with GPT-3, in our main results, we sample a subset (See Appendix C) from the original test set for two datasets: ACE05 and TACRED as shown in Table 1.

**Semeval 2010 task 8** Hendrickx et al. (2010) focuses on semantic relations between pairs of nominals collected from general domain resources.

**TACRED** Zhang et al. (2017) is a large-scale relation extraction dataset with 106,264 examples built over newswire and web text.

**SciERC** Luan et al. (2018) collects AI paper abstracts and annotated relations, especially for scientific knowledge graph construction.

**ACE05** contains the entity, relation, and event annotations collected from domains including newswire, broadcast, discussion forums, etc.

### 3.2 Baseline Methods

**GPT-3 baselines** For GPT-3 baselines and our methods, we select "text-davinci-003" with maximal 4,097 input tokens and use the identical prompt construction (Sec. 2.3) via OpenAI API. We implement two categories of GPT-3 baselines:

**(1) GPT-Random** Instead of randomly selecting few-shot demonstrations from the training data for each test input, we add extra constraints to make the label distribution of selected demonstrations more uniform. Our preliminary experiments suggest that this is a stronger baseline than the vanilla random.

**(2) GPT-Sent** Previous work attempts various sentence embedding in retrieval. In this work, our implementation adopted SimCSE (Gao et al., 2021), which has been demonstrated to be the state-of-the-art method for sentence similarity tasks.

**Fine-tuned RE Models** In our experiment, we choose PURE (Zhong and Chen, 2021), an entity marker-based fine-tuned model mentioned in Sec. 2.4.2 to obtain the representations for retrieval. Meanwhile, PURE performs as a directly comparable baseline. We also compare with corresponding SOTA fine-tuned baselines on Semeval Cohen et al. (2020) (reformulate RE as the question answering task) and TACRED Wang et al. (2022a) (extra pre-training to capture RE structure) datasets.

All implementation details are in Appendix A.

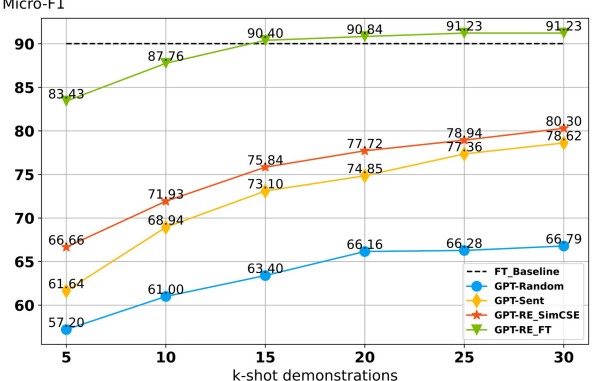

(a) **The comparison on retrieval modules**

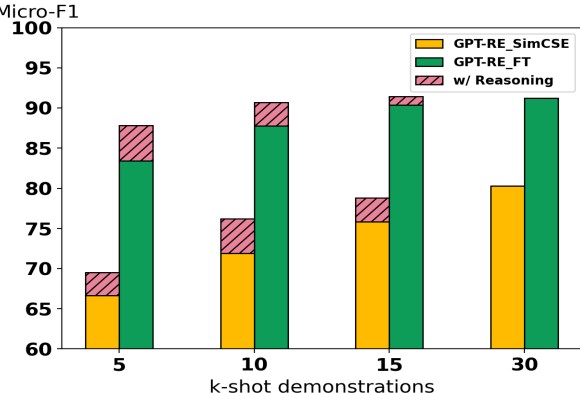

(b) **Reasoning with fewer demonstrations**.

Figure 6: **Ablation study on the retrieval and reasoning components on Semeval.** We sampled a subset from the test data with 300 examples. We show the 'w/o reasoning' results with $k = 30$ for comparison.

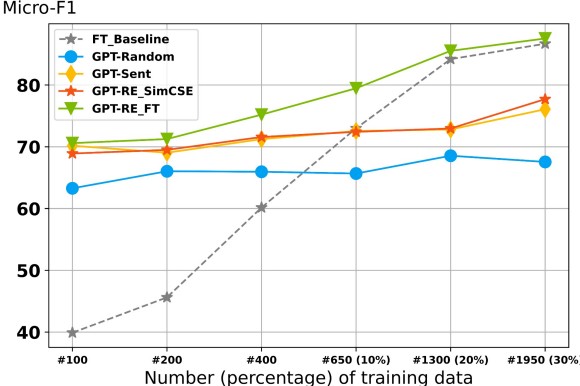

Figure 7: **Low-resource Scenario on Semeval**. We limit the percentage of training data for both fine-tuning and retrieval in GPT-RE.

of demonstrations. Meanwhile, the high-quality demonstrations obtained by *GPT-RE_FT* offset the effort of enriching reasoning into demonstrations, which shows relatively trivial improvements. Since reasoning aims at enriching demonstrations, this feature potentially works better with fewer demonstrations, as shown in Section 4.3.

## 4.2 Ablation Study on Task-aware Retrieval

We first implement the ablation experiments of the retrieval component with the setting of increasing $k$-shot demonstrations (Figure 6a). We find that: (1) compared to *GPT-Random*, all the retrieval-based models have higher F1 scores and large gradients of the performance curves. It means that GPT-3 can learn from high-quality demonstrations more effectively; (2) after adding entity information to the SimCSE retrieval, *GPT-RE_SimCSE* achieves better performance throughout all $K$ shots, indicating that task-aware sentence embedding can capture the feature of RE and provide more proper demonstrations; (3) finally, the fine-tuned relation representation retriever *GPT-RE_FT* significantly outperforms all retrieval-based methods and beats the fine-tuning baseline when $k > 15$. Note that even with $k = 5$ demonstrations, *GPT-RE_FT* still works better than *GPT-RE_SimCSE* with $k = 30$ ($80.30 \rightarrow 83.43(+3.13)$), which indicates that the quality of demonstrations shows much more important than the number of demonstrations.

## 4.3 Ablation Study on Reasoning Enhancing

We then check the influence of our proposed reasoning-enhanced demonstration, as shown in Figure 6b. Due to the limited amount of input tokens of GPT-3, we have to set the $k \leq 15$ for the

# 4 Experimental Results

## 4.1 Main Results

We compare our main experiment results with previous methods in Table 2. **GPT-RE_SimCSE** denotes our entity-prompted sentence embedding for retrieval and **GPT-RE_FT** denotes our fine-tuned relation representation for retrieval. From the table, we can observe that: (1) both *GPT-RE_SimCSE* and *GPT-RE_FT* outperform the retrieval-based *GPT-Sent*, indicating that it is necessary to inject the task-specific information into sentence embedding for selecting proper demonstrations; (2) *GPT-RE_FT* succeeds to outperform the fine-tuning baseline PURE on three datasets by $+2.00$, $+2.42$, $+0.55$ Micro-F1. It suggests that GPT-3 has the potential to beat fine-tuning when the retriever has prior task knowledge. *GPT-RE_FT* eventually achieves SOTA results on Semeval and SciERC. (3) reasoning module improves *GPT-RE_SimCSE* by around $2\%$ Micro-F1, indicating that gold label-induced reasoning successfully enriches the knowledge

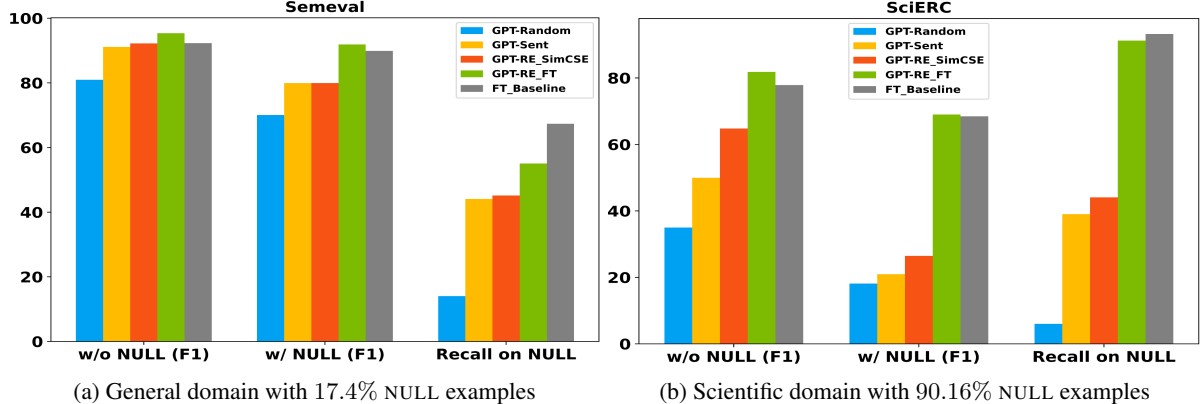

(a) General domain with 17.4% NULL examples       (b) Scientific domain with 90.16% NULL examples

Figure 8: **Analysis on the effects of NULL examples**. w/o NULL refers to the classification setting that NULL examples are excluded from the train and test data. w/ NULL refers to the original extraction setting. We use the full test set for the evaluation.

tokens of reasoning, leading to a trade-off between adding reasoning and adding more demonstrations. From the result, we find that: (1) with reasoning-enhanced demonstrations, GPT-3 always achieves better scores across all the $k$-shot settings of both *GPT-RE_SimCSE* and *GPT-RE_FT*, indicating that the reasoning induced from ground truth relation labels can effectively unlock the reasoning ability of GPT-3 and improve the ICL with a deeper understanding of demonstrations. Specifically, for *GPT-RE_FT*, the performance improvement becomes less significant when more demonstrations are provided, which is feasible as with more high-quality demonstrations available, GPT-3 can already learn the internal reasoning behind each demonstration; (2) since the reasoning enhancement works better with fewer demonstrations, we expect this method can be an effective solution to low-shot relation extraction (Han et al., 2018; Geng et al., 2020; Liu et al., 2022a), which aims at recognizing novel relations with very few or no examples, and we leave this for future work.

### 4.4 Low-resource Scenario

We conduct the experiment for observing the low-resource performance in the general domain Semeval task. As shown in Figure 7, we observe that: (1) all the GPT-3 based results work better than fine-tuning in when the training examples are less than # 650 (10%). It indicates that in the general domain RE, GPT-3 benefits from its abundant prior knowledge to understand the relations; (2) *GPT-RE_SimCSE* starts to show a substantial difference to *GPT-Sent* after the training size surpasses 30%. We believe fewer training candidates

could limit the effects of retrieval; (3) *GPT-RE_FT* achieves an upper bound performance in all settings, even when the fine-tuned model shows poor performance with hundreds of training data (from #100 to #400). This emphasizes the impressive effectiveness of fine-tuned relation representations for capturing higher-quality demonstrations. The observation in the low-resource setting is very different from Gutiérrez et al. (2022). We assume the difference could be caused by the domain and NULL proportion of the task.

## 5 Analysis

### 5.1 The Issue of "Overpredicting"

To analyze the influence of NULL class, we compare the effectiveness of each method for alleviating this issue on two datasets: general domain Semeval with 17.4% NULL examples and scientific domain SciERC with 90.16% NULL examples. As shown in Figure 8, (1) by comparing the performance on Semeval and SciERC, a larger percentage of NULL examples results in more significant performance drop showing the negative influence of overpredicting NULL examples; (2) by comparing w/o NULL and w/ NULL, our *GPT-RE_FT* shows the most robustness to the influence of NULL examples, indicating that the RE fine-tuned representations in retrieval can release the overpredicting issue of GPT-3 by providing higher-quality demonstrations; (3) however, even with task-aware representations, all GPT-3 methods still underperform the fine-tuning baseline on NULL examples, this is due to the confusing definition of NULL, in many cases, there is a certain relation between entities in the context, but out of the distribution of pre-

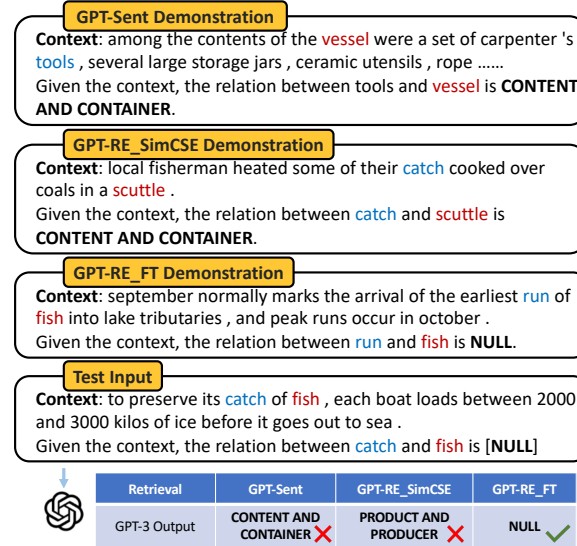

**GPT-Sent Demonstration**
**Context**: among the contents of the vessel were a set of carpenter 's tools , several large storage jars , ceramic utensils , rope ......
Given the context, the relation between tools and vessel is **CONTENT AND CONTAINER**.

**GPT-RE_SimCSE Demonstration**
**Context**: local fisherman heated some of their catch cooked over coals in a scuttle .
Given the context, the relation between catch and scuttle is **CONTENT AND CONTAINER**.

**GPT-RE_FT Demonstration**
**Context**: september normally marks the arrival of the earliest run of fish into lake tributaries , and peak runs occur in october .
Given the context, the relation between run and fish is **NULL**.

**Test Input**
**Context**: to preserve its catch of fish , each boat loads between 2000 and 3000 kilos of ice before it goes out to sea .
Given the context, the relation between catch and fish is **[NULL]**

| Retrieval | GPT-Sent | GPT-RE_SimCSE | GPT-RE_FT |
|-----------|----------|---------------|-----------|
| GPT-3 Output | CONTENT AND CONTAINER ✗ | PRODUCT AND PRODUCER ✗ | NULL ✓ |

Figure 9: **A case study of demonstration quality on Semeval.** [NULL] is the gold label here.

defined classes. In these cases, GPT-3 tends to overpredict as the relation information may be covered in its prior knowledge. We think this ability of GPT-3 can be useful in more open fields, such as open RE (Banko and Etzioni, 2008) which has no pre-defined relation classes.

## 5.2 Case Study of Demonstration Quality

We select one typical test example to better illustrate the amendment of our task-aware demonstration retrieval. As shown in Figure 9, given the NULL Example, we show the most similar demonstration in retrieval based on three methods. The *GPT-Sent* retrieved demonstration focuses on the semantic meaning of "CONTENT AND CONTAINER" which is shared in the test context, but not revealed in the target entity pair. This mismatch confirms the problem of lacking entity information in retrieval. Instead, *GPT-RE_SimCSE* retrieves a much more relevant demonstration that shows the same semantic relation between "catch" and "fish" but still faces a minor mismatch as the gold label is between "catch" and "scuttle." Finally, *GPT-RE_FT* demonstration shares a similar structure with the test input regarding the pair of entities, which is the key clue for predicting the relation between entities. This result shows a level-by-level enhancement with more entity information provided in retrieval. We also show some other case examples in Appendix B.

## 6 Related Work

**In-context Learning** Recent work shows that ICL of GPT-3 (Brown et al., 2020) can perform numerous tasks when provided a few examples in a natural language prompt. Existing work focuses on various aspects to effectively utilize the advantages of GPT-3, from prompt design (Perez et al., 2021) for proper input to coherence calibration (Malkin et al., 2022) for tackling the diverse generated output. Another research path locates in the demonstration part, including ordered prompts (Lu et al., 2022) and retrieval-based demonstrations (Rubin et al., 2022; Liu et al., 2022b; Shin et al., 2021).

To the best of our knowledge, there is no previous work exploring the potential of GPT-3 on general domain RE tasks. A recent work attempts to leverage GPT-3 in biomedical information extraction (NER and RE), and reveals issues of ICL that may be detrimental to IE tasks in general. Our work succeeds in overcoming these issues to some extent and confirms the potential of GPT-3 in both general and the scientific domain RE.

**Retrieval-based Demonstrations** Several studies have demonstrated that dynamically selecting few-shot demonstrations for each test example, instead of utilizing a fixed set, leads to significant improvement in GPT-3 ICL (Liu et al., 2022b; Shin et al., 2021; Rubin et al., 2022). They also show that nearest neighbor in-context examples yield much better results than the farthest ones. This leads to the significance of better retrieval modules for demonstrations. Existing attempts rely on sentence embedding in retrieval, including the sentence encoders of PLMs such as BERT (Devlin et al., 2019), RoBERTa (Zhuang et al., 2021) KATE (Liu et al., 2022b) , SimCSE (Gao et al., 2021), Sentence-BERT (Reimers and Gurevych, 2019; Wolf et al., 2020). Unlike these sentence embeddings, we propose to fine-tune PLMs on our target RE tasks to produce more task-specific and robust representations for retrieval.

## 7 Conclusions

This work explores the potential of GPT-3 ICL on RE for bridging the performance gap to the fine-tuning baselines via two strategies: (1) task-aware demonstration retrieval emphasizes entity and relation information for improving the accuracy of searching demonstrations; (2) gold label-induced reasoning enriches the reasoning evidence

of each demonstration. To the best of our knowledge, GPT-RE is the first GPT-3 ICL research that significantly outperforms the fine-tuning baseline on three datasets and achieves SOTA on Semeval and SciERC. We implement detailed studies to explore how GPT-3 overcomes the difficulties such as NULL example influence.

## Limitations

Despite the overall positive results, GPT-RE still faces two shortcomings: (1) the issue of overpredicting has been significantly alleviated but not completely solved, and the NULL recall still lags behind full-supervised baselines, especially on the datasets containing a large proportion of NULL examples such as ACE05 ("95.60%"); (2) Though the task-aware retriever optimizes the representations of PLMs such as SimCSE and BERT, it is widely considered that LLMs can generate more robust representations than small PLMs. Future work can replace representations generated by smaller PLMs with GPT-3 itself. However, due to the access limitation to the representations of GPT-3, we can nearly confirm this proposal up to now.

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

| Hyperparameter | In Experiment |
|---|---|
| Engine | text-davinci-003 |
| Temperature | 0.0 |
| Max_tokens | 256 |
| Top_p | 1 |
| Frequency_penalty | 0.0 |
| Presence_penalty | 0.0 |
| Best_of | 1 |
| Logprob | 1 |

Table 3: GPT-3 Hyperparamters.

| Dataset | Lower bound | Upper bound |
|---|---|---|
| Semeval | 5 | 30 |
| TACRED | 5 | 15 |
| SciERC | 5 | 30 |
| ACE05 | 5 | 25 |

Table 4: Search range for each dataset.

# A Hyperparameters

## A.1 GPT-3 Hyperparameters

We use the GPT-3 API during the experiments and set the hyperparameters as in Table 3. Since the "Temperature" is set to be 0.0, denoting the stable output of GPT-3, we report the result of the single run for all experiments. Due to the input length limitation of GPT-3 and the various average lengths of contexts from each dataset, we set different search ranges for the number of demonstrations of each dataset as shown in Table 4.

## A.2 Fine-tuning Baseline PURE

We follow their single-sentence setup to keep consistency among datasets as Semeval and TACRED are both sentence-level RE datasets. For the PLMs, we also follow PURE by using *scibert-scivocab-uncased* (Beltagy et al., 2019) as the base encoder for SciERC and *bert-base-uncased* (Devlin et al., 2019) for the remaining three general domain datasets. We follow hyperparameters in their paper. We used 2 NVIDIA RTX3090 for training.

## A.3 Sentence Embedding Methods

Gutiérrez et al. (2022) uses the [CLS] of RoBERTa-large as the representation in retrieval, Liu et al. (2022b) fine-tunes RoBERTa-large on two natural language inference (NLI) datasets: SNLI (Bowman et al., 2015) and MultiNLI (Williams et al., 2018) to enhance the quality of sentence embedding. For the sentence embedding method SimCSE in our experiment, we utilize the version: sup-simcse-bert-base-uncased.

**GPT-Sent Demonstration**
**Context**: this paper describes a set of principles designed to help archives position themselves to address the management ......
Given the context, the relation between principles and set is **MEMBER AND COLLECTION**.

**GPT-RE_SimCSE Demonstration**
**Context**: the screen works using ink , just like books and newspapers , but displays the ink particles electronically .
Given the context, the relation between ink and screen is **COMPONENT AND WHOLE**.

**GPT-RE_FT Demonstration**
**Context**: the computer mouse has been the input device of choice for a long time now in the computer world .
Given the context, the relation between mouse and computer is **COMPONENT AND WHOLE**.

**Test Input**
**Context**: basic diagrams also work well on the computer screen if they are carefully designed to match the grid of pixels on the screen.
Given the context, the relation between screen and computer is [**COMPONENT AND WHOLE**]

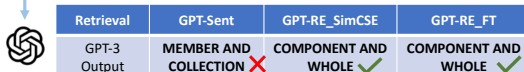

| Retrieval | GPT-Sent | GPT-RE_SimCSE | GPT-RE_FT |
|---|---|---|---|
| GPT-3 Output | MEMBER AND COLLECTION ✗ | COMPONENT AND WHOLE ✓ | COMPONENT AND WHOLE ✓ |

(a) [**COMPONENT AND WHOLE**] denotes the gold label

**GPT-Sent Demonstration**
**Context**: a woman diagnosed with breast cancer today joins a huge sisterhood of cancer survivors ready to help her along the way ......
Given the context, the relation between survivors and sisterhood is **MEMBER AND COLLECTION**.

**GPT-RE_SimCSE Demonstration**
**Context**: the victim of last night 's car accident donated his organs to several patients who have been waiting for donated organs .
Given the context, the relation between organs and patients is **ENTITY AND DESTINATION**.

**GPT-RE_FT Demonstration**
Context: operation homefront and partners delivered toys to military children .
Given the context, the relation between toys and children is **ENTITY AND DESTINATION**.

**Test Input**
Context: the wheelchair foundation donated wheelchairs to people with physical problems in hundred countries .
Given the context, the relation between wheelchairs and people is [**ENTITY AND DESTINATION**].

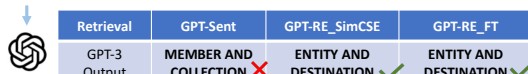

| Retrieval | GPT-Sent | GPT-RE_SimCSE | GPT-RE_FT |
|---|---|---|---|
| GPT-3 Output | MEMBER AND COLLECTION ✗ | ENTITY AND DESTINATION ✓ | ENTITY AND DESTINATION ✓ |

(b) [**ENTITY AND DESTINATION**] denotes the gold label.

Figure 10: **More casees.**

| Label | # Num |
|---|---|
| PHYS | 28 |
| GEN-AFF | 12 |
| PER-SOC | 11 |
| GEN-AFF | 33 |
| PART-WHOLE | 13 |
| ART | 19 |
| NULL | 2329 |

Table 5: ACE05

## B Case Study

To verify the effectiveness of our task-aware demonstration retrieval, we provide more cases.

For Figure 10a, *GPT-Sent* retrieves a demonstration that shares the same semantic meaning of "design" with the test input. However, the entity pair is irrelevant to the concept "design" resulting in a noisy demonstration. Instead, *GPT-RE_SimCSE* retrieves a more relative demonstration with closer pair of entities sharing the same relation label. Furthermore, *GPT-RE_FT* retrieves the demonstration containing both the closing entity pair and the same linguistic structure between entities. This case emphasizes level-by-level improvement using our proposed methods. Figure 10b shows a similar phenomenon.

## C Subset

The number of sampled examples is not only related to the size of the training data itself. A more important factor is the proportion of NULL. We have to maintain the original label distribution in datasets with a high proportion of NULL. Thus, the rule to sample the subset is to keep the proportion of each relation label consistent with the original test set. Table 5 6 are label distributions of two subsets.

GPT-RE_FT on TACRED surpasses the supervised baseline in the current subset. As we show above, some labels in TACRED are indeed not well presented (only 1 example), since TACRED dataset contains some long-tail labels. We decided to add additional results of GPT-RE_FT by enlarging our sampled set to # 3200 (2 times the current version), and the performance of GPT-RE_FT ($k = 15$) is 73.16 while the performance of PURE is 70.48.

| Label | # Num |
|---|---|
| Per:title | 40 |
| PER:city_of_death | 1 |
| Org:shareholders | 2 |
| Per:origin | 12 |
| Org:top_members/employees | 36 |
| Org:city_of_headquarters | 11 |
| Per:religion | 4 |
| Per:city_of_birth | 1 |
| Per:employee_of | 27 |
| Per:data_of_death | 3 |
| Per:other_family | 5 |
| Org:website | 6 |
| Per:cause_of_death | 3 |
| Org:subsidiaries | 4 |
| Org:stateorprovince_of_headquarters | 5 |
| Per:countries_of_residence | 10 |
| Per:siblings | 5 |
| Per:stateorprovinces_of_residence | 11 |
| Org:alternate_names | 27 |
| Per:spouse | 4 |
| Per:parents | 7 |
| Org:country_of_headquarters | 9 |
| Per:age | 21 |
| Per:date_of_birth | 1 |
| Per:country_of_death | 1 |
| Per:schools_attended | 4 |
| Org:member_of | 3 |
| Per:children | 5 |
| Org:parents | 7 |
| Per:cities_of_residence | 24 |
| Per:stateorprovince_of_brith | 1 |
| Per:charges | 12 |
| Org:founded | 2 |
| Org:country_founded_by | 5 |
| Per:stateorprovince_of_death | 1 |
| Org:members | 4 |
| Per:country_of_birth | 1 |
| Per:alternate_names | 1 |
| Org:number_of_employees/members | 1 |
| Org:dissolved | 1 |
| Org:political/religious_affiliation | 1 |
| NULL | 1271 |

Table 6: TACRED