# OpenReview forum: "GPT-RE: In-context Learning for Relation Extraction using Large Language Models"
_EMNLP/2023/Conference — EMNLP 2023 Main_

### Official Review · Reviewer_wnQk · 2023-07-20

**Soundness:** 4

**Excitement:**

4: Strong: This paper deepens the understanding of some phenomenon or lowers the barriers to an existing research direction.

**Paper Topic And Main Contributions:**

The authors present an approach for the relation extraction task (RE) using large language models (LLM) and via in-context learning (ICL). They address the current shortcomings of ICL for RE, which according to the authors are the reason for poor results of LLMs used for RE. The authors try to solve the problem by (1) incorporating task-aware representations in demonstration retrieval (i.e. representations emphasizing entity and relation instead of kNN search based on sentence embeddings) and (2) enriching the demonstrations with gold label-induced reasoning logic. Additionally the problem of overpredicting of NULL examples is analyzed.

The approach is tested on four RE datasets (Semeval, TACRED, SciERC, ACE05). The approach is compared to two GPT-3 baselines (one using randomly selecting the demonstrations and another using sentence embedding retrieval) as well as fully-supervised models. The new model GPT-RE achieves SOTA performances on the Semeval and SciERC datasets, and competitive performance on TACRED and ACE05 datasets.

The authors analyze the results on multiple levels: (1) An ablation study regarding the retrieval of an increasing number of demonstrations shows that task-aware retrieval beats a random selection as well as the sentence embedding approach even for the lowest k demonstrations which indicates that the quality of demonstrations is more important than the number of demonstrations. (2) An ablation study on reasoning enhancing shows that inducing reasoning improves the performance, especially with fewer demonstrations, which might be an effective solution for low-shot relation extraction (3) A comparison of the performance in a setting without NULL examples shows that GPT-RE_FT (GPT with fine-tuned relation representation for retrieval) is most robust to the influence of a number of NULL examples, however it underperforms the fine-tuning baseline on NULL examples.

**Questions For The Authors:**

Question A: Can you provide more information on the distribution of the relations in test subsets ?

**Reasons To Accept:**

- The paper is well organized and mostly easy to follow.
- The study addresses a performance gap between fine-tuned models and in-context learning for relation extraction, and it proposes a solution which achieve SoTA or at least competitive results on multiple, widely used RE datasets.
- In additional ablation studies a deeper analysis of the performance demonstrates the potential of the approach in different settings.
- Overall the study is an interesting contribution to the relation extraction task

**Reasons To Reject:**

Regarding the experiments done on TACRED and ACE05: due to the cost of running the model the authors sampled test subsets from that data, which is larger then SEMEVAL and SciERC. The approach is understandable. However, strangely the sampled subsets are smaller than the whole test data of the other two datasets; I would expect the subsets of TACRED and ACE05 to have at least the same size or even a bigger size as the other two datasets in order to reflect the whole test data better. It does hamper the comparison between the models. This is especially to relevant wrt TACRED, which contains a much larger number of relations than other datasets, and  many of them are probably not well represented in the sampled subset. The authors plan to release the subsets, however the paper contains no information about the distribution of the relations in the test subsets.

**Reproducibility:**

4: Could mostly reproduce the results, but there may be some variation because of sample variance or minor variations in their interpretation of the protocol or method.

**Reviewer Confidence:**

3: Pretty sure, but there's a chance I missed something. Although I have a good feel for this area in general, I did not carefully check the paper's details, e.g., the math, experimental design, or novelty.

**Typos Grammar Style And Presentation Improvements:**

- l. 333: retrieval

---

> ### Author Rebuttal · Authors · 2023-08-29
>
> Thanks for pointing out the typos, and we appreciate your valuable advice for improving this paper.
>
> **Question**: "Can you provide more information on the distribution of the relations in test subsets ?"
>
> **Answer**: The number of sampled examples is not only related to the size of the training data itself. A more important factor is the proportion of NULL. In datasets with a high proportion of NULL, we have to maintain the original label distribution. Thus, the rule to sample the subset is to keep the proportion of each relation label consistent with the original test set.
> Below are label distributions of two subsets.
>
> For ACE05,
> | Label |  #num|
> | ----------- | ----------- |
> | PHYS |  26|
> | GEN-AFF |  12|
> | PER-SOC |  11|
> | ORG-AFF |  33|
> | PART-WHOLE |  13|
> | ART |  19|
> | NULL |  2329|
>
> For TACRED,
> | Label |  #num|
> | ----------- | ----------- |
> |Per:title | 40|
> |Per:city_of_death | 1|
> |Org”shareholders | 2|
> |Per:origin | 12|
> |Org:top_members/employees | 36|
> |Org:city_of_headquarters | 11|
> |Per:religion | 4|
> |Per:city_of_birth | 1|
> |Per:employee_of | 27|
> |Per:data_of_death | 3|
> |Per:other_family | 5|
> |Org:website | 6|
> |Per:cause_of_death | 3|
> |Org:subsidiaries | 4|
> |Org:stateorprovince_of_headquarters | 5|
> |Per:countries_of_residence | 10|
> |Per:siblings | 5|
> |Per:stateorprovinces_of_residence | 11|
> |Org:alternate_names | 27|
> |Per:spouse | 4|
> |Per:parents | 7|
> |Org:country_of_headquarters | 9|
> |Per:age | 21|
> |Per:date_of_birth | 1|
> |Per:country_of_death | 1|
> |Per:schools_attended | 4|
> |Org:member_of | 3|
> |Per:children | 5|
> |Org:parents | 7|
> |Per:cities_of_residence | 24|
> |Per:stateorprovince_of_brith | 1|
> |Per:charges | 12|
> |Org:founded | 2|
> |Org:country_founded_by | 5|
> |Per:stateorprovince_of_death | 1|
> |Org:members | 4|
> |Per:country_of_birth | 1|
> |Per:alternate_names | 1|
> |Org:number_of_employees/members | 1|
> |Org:dissolved | 1|
> |Org:political/religious_affiliation | 1|
> |NULL | 1271|
>
> We also think it important to choose a reasonable sample strategy for these two datasets.
> - For comparison with previous GPT-3 baselines, since the performance gap is too large between previous GPT-3 baselines and our GPT-RE, we think that the current subset is sufficient to show the effectiveness. Also note that some previous GPT3-based work uses much smaller subsets in their experiment (e.g., #1000 examples for each dataset in refer [1], and we have tried our best to choose as large as possible proportion considering our allowing.
> - For comparison with the SOTA supervised baselines,
>     - GPT-RE_FT on ACE05 achieves competitive performance since we keep the same proportion of NULL in our subset, otherwise, the GPT-RE may outperform the supervised baseline. We did experiments by removing all NULL examples in the test set, leading to the improvement of all GPT-based methods, which is consistent with Figure 8.
>     - GPT-RE_FT on TACRED surpasses the supervised baseline in the current subset. As we show above, some labels in TACRED are indeed not well presented (only 1 example), since TACRED dataset contains some long-tail labels. We decided to add additional results of GPT-RE_FT by enlarging our sampled set to # 3200 (2 times the current version). We already finished the experiment, and the performance of GPT-RE_FT (k = 15) is 73.16 while the performance of PURE is 70.48.
>
> We will add statistics of subsets above, and the additional experiment results in our camera-ready version.
>
> [1]: Thinking about GPT-3 In-Context Learning for Biomedical IE? Think Again. (Gutiérrez +, EMNLP 2022 Findings)

---

### Official Review · Reviewer_Ucne · 2023-08-05

**Soundness:** 4

**Excitement:**

4: Strong: This paper deepens the understanding of some phenomenon or lowers the barriers to an existing research direction.

**Paper Topic And Main Contributions:**

In this paper, the authors proposed an approach for relation extraction. Grounded on large language models (LLMs) via in-context learning (ICL), it prompts LLMs with the task instruction and demonstration data to generate the corresponding relation. They proposed two methods for representation, task-aware representations and  gold label-induced reasoning. The first includes Entity-Prompted Sentence Embedding, which reconstructs the context for entity information and extracting relation representation from a fine-tuned RE model and use these representations for retrieval. Experiments were done on four datasets: Semeval, TACRED, SciERC, and ACE05. The proposed model achieves state-of-the-art performance on the Semeval and SciERC datasets.

The paper is well written with good survey of related works.

The authors showed that with task-aware representations, all the retrieval-based models have higher performance than the GPT-Random, while the gold label-induced reasoning helps to improve the performance.

The approach can be a good resource for future research of LLMs to resolve the RE problems.

**Questions For The Authors:**

In section 2.4.2, we define the relation representation as Rel = hi + hj. Does the relation representation include the context in between, or does it include only the representations of two entities? If it is the former, it is not surprising that the GPT-RE_FT model achieves good performance, the way the relation representation is obtained is similar to previous works, but instead of putting through a neural network, now it is put into a retriever. The GPT-RE_SimCSE has the novelty in the entity prompts, which is still limited.

However, if it is the latter, as stated in section 2.4.2 "representation Rel is naturally enriched with the entity information.", it is kind of weird that the GPT-RE_FT can achieves state-of-the-art with only entity information and without context.

In section 3.1, "Due to the cost of running the model in the API with GPT-3, in our main results, we sample a subset". What are the rules to take the proportion for the subset? It seems the subset size is not proportional to either the size of training or test set.

**Reasons To Accept:**

The paper is well written with good survey of related works.

The authors showed that with task-aware representations, all the retrieval-based models have higher performance than the GPT-Random, while the gold label-induced reasoning helps to improve the performance.

The approach can be a good resource for future research of LLMs to resolve the RE problems.

**Reasons To Reject:**

In section 2.4.2, we define the relation representation as Rel = hi + hj. Does the relation representation include the context in between, or does it include only the representations of two entities? If it is the former, it is not surprising that the GPT-RE_FT model achieves good performance, the way the relation representation is obtained is similar to previous works, but instead of putting through a neural network, now it is put into a retriever. The GPT-RE_SimCSE has the novelty in the entity prompts, which is still limited.

However, if it is the latter, as stated in section 2.4.2 "representation Rel is naturally enriched with the entity information.", it is kind of weird that the GPT-RE_FT can achieves state-of-the-art with only entity information and without context.

**Reproducibility:**

4: Could mostly reproduce the results, but there may be some variation because of sample variance or minor variations in their interpretation of the protocol or method.

**Reviewer Confidence:**

4: Quite sure. I tried to check the important points carefully. It's unlikely, though conceivable, that I missed something that should affect my ratings.

---

> ### Author Rebuttal · Authors · 2023-08-29
>
> Thank you for your valuable advice for improving this paper.
>
> **Question**: “In section 2.4.2, we define the relation representation as Rel = hi + hj. Does the relation representation include the context in between, or does it include only the representations of two entities? If it is the former, it is not surprising that the GPT-RE_FT model achieves good performance, the way the relation representation is obtained is similar to previous works, but instead of putting it through a neural network, now it is put into a retriever. The GPT-RE_SimCSE has the novelty in the entity prompts, which is still limited.
> However, if it is the latter, as stated in section 2.4.2 "representation Rel is naturally enriched with the entity information.", it is kind of weird that the GPT-RE_FT can achieve state-of-the-art with only entity information and without context.
>
> **Answer**: In format, “Rel = hi + hj” is the concatenation of two marker vectors regarding two entities and does not include the surrounding vectors in between. However,  transformer-based pre-trained models like BERT naturally learn contextualized token representations that aggregate context information by the self-attention mechanism.  During the fine-tuning of BERT-based RE models, the representation “Rel” has already been trained to capture the relevant contextual information to predict the target label, and existing work (refer [1][2]) has proved the SOTA performance of marker-based methods in RE. Thus, semantically we could say that “Rel” includes the context between two entities.
>
> [1]: A Frustratingly Easy Approach for Entity and Relation Extraction. (Zhong +, NAACL 2021)
>
> [2]: Matching the Blanks: Distributional Similarity for Relation Learning. (Soares +, ACL 2019)
>
> **Question**: In section 3.1, "Due to the cost of running the model in the API with GPT-3, in our main results, we sample a subset". What are the rules to take the proportion for the subset? It seems the subset size is not proportional to either the size of training or test set.
>
> **Answer**: The number of sampled examples is not only related to the size of the training data itself. A more important factor is the proportion of NULL. In datasets with a high proportion of NULL, we have to maintain the original label distribution. Thus, the rule to sample the subset is to keep the proportion of each relation label consistent with the original test set.
> Below are label distributions of two subsets.
>
> For ACE05,
> | Label |  #num|
> | ----------- | ----------- |
> | PHYS |  26|
> | GEN-AFF |  12|
> | PER-SOC |  11|
> | ORG-AFF |  33|
> | PART-WHOLE |  13|
> | ART |  19|
> | NULL |  2329|
>
> For TACRED,
> | Label |  #num|
> | ----------- | ----------- |
> |Per:title | 40|
> |Per:city_of_death | 1|
> |Org”shareholders | 2|
> |Per:origin | 12|
> |Org:top_members/employees | 36|
> |Org:city_of_headquarters | 11|
> |Per:religion | 4|
> |Per:city_of_birth | 1|
> |Per:employee_of | 27|
> |Per:data_of_death | 3|
> |Per:other_family | 5|
> |Org:website | 6|
> |Per:cause_of_death | 3|
> |Org:subsidiaries | 4|
> |Org:stateorprovince_of_headquarters | 5|
> |Per:countries_of_residence | 10|
> |Per:siblings | 5|
> |Per:stateorprovinces_of_residence | 11|
> |Org:alternate_names | 27|
> |Per:spouse | 4|
> |Per:parents | 7|
> |Org:country_of_headquarters | 9|
> |Per:age | 21|
> |Per:date_of_birth | 1|
> |Per:country_of_death | 1|
> |Per:schools_attended | 4|
> |Org:member_of | 3|
> |Per:children | 5|
> |Org:parents | 7|
> |Per:cities_of_residence | 24|
> |Per:stateorprovince_of_brith | 1|
> |Per:charges | 12|
> |Org:founded | 2|
> |Org:country_founded_by | 5|
> |Per:stateorprovince_of_death | 1|
> |Org:members | 4|
> |Per:country_of_birth | 1|
> |Per:alternate_names | 1|
> |Org:number_of_employees/members | 1|
> |Org:dissolved | 1|
> |Org:political/religious_affiliation | 1|
> |NULL | 1271|
>
> We also think it important to choose a reasonable sample strategy for these two datasets.
> - For comparison with previous GPT-3 baselines, since the performance gap is too large between previous GPT-3 baselines and our GPT-RE, we think that the current subset is sufficient to show the effectiveness. Also note that some previous GPT3-based work uses much smaller subsets in their experiment (e.g., #1000 examples for each dataset in refer [1], and we have tried our best to choose as large as possible proportion considering our allowing.
> - For comparison with the SOTA supervised baselines,
>     - GPT-RE_FT on ACE05 achieves competitive performance since we keep the same proportion of NULL in our subset, otherwise, the GPT-RE may outperform the supervised baseline. We did experiments by removing all NULL examples in the test set, leading to the improvement of all GPT-based methods, which is consistent with Figure 8.
>     - GPT-RE_FT on TACRED surpasses the supervised baseline in the current subset. As we show above, some labels in TACRED are indeed not well presented (only 1 example), since TACRED dataset contains some long-tail labels. We decided to add additional results of GPT-RE_FT by enlarging our sampled set to # 3200 (2 times the current version). We already finished the experiment, and the performance of GPT-RE_FT (k = 15) is 73.16 while the performance of PURE is 70.48.
>
> We will add statistics of subsets above, and the additional experiment results in our camera-ready version.
>
> [1]: Thinking about GPT-3 In-Context Learning for Biomedical IE? Think Again. (Gutiérrez +, EMNLP 2022 Findings)

---

### Official Review · Reviewer_UZWY · 2023-08-10

**Soundness:** 4

**Excitement:**

4: Strong: This paper deepens the understanding of some phenomenon or lowers the barriers to an existing research direction.

**Paper Topic And Main Contributions:**

The paper underscores potential limitations associated with in-context learning (ICL) and suggests that these limitations might be the contributing factor to the underperformance of large language models (LLMs) with ICL when compared to fully supervised baselines in the context of relation extraction (RE). The identified shortcomings encompass two key aspects "(1) low relevance regarding entity and relation in existing sentence-level demonstration retrieval approaches for ICL; and (2) the lack of explaining input-label mappings of demonstrations leading to poor ICL effectiveness."
Within the scope of this paper, the authors introduce their GPT-RE model as a solution to these challenges. Notably, the paper also highlights a pitfall observed in applying GPT-3's ICL approach to RE. To counter these issues, GPT-RE employs two strategic approaches: (1) task-aware retrieval and (2) gold label-induced reasoning. Moreover, the paper delves into addressing a newly identified concern termed "overpredicting."
In the experimental phase, the authors demonstrate that the GPT-RE approach not only surpasses GPT-3 but also outperforms fully-supervised baselines. This highlights the efficacy of their model in enhancing RE performance.








**Reasons To Accept:**

The paper adeptly highlighted a potential concern regarding GPT-3's utilization of in-context learning (ICL) and effectively provided a resolution. The observed performance enhancements are indeed promising, as they notably elevate the performance beyond that of GPT-3 alone. This achievement suggests that the strategies introduced in the paper hold the potential to offer benefits to a wide array of large language models (LLMs).

**Reasons To Reject:**

I think the paper could further strengthen its argument by providing additional evidence to substantiate the identified shortcomings of in-context learning (ICL). By delving deeper into the reasons behind these shortcomings, the paper could enhance its overall credibility and contribute to a more comprehensive understanding of the challenges associated with ICL.

**Reproducibility:**

5: Could easily reproduce the results.

**Reviewer Confidence:**

3: Pretty sure, but there's a chance I missed something. Although I have a good feel for this area in general, I did not carefully check the paper's details, e.g., the math, experimental design, or novelty.

---

> ### Author Rebuttal · Authors · 2023-08-29
>
> Thanks for your valuable advice for improving this paper.
> To deepen the understanding of both the shortcomings of ICL and the mechanism behind our proposal to improve ICL, we think it necessary to get deeper access to the GPT-3 models (e.g., embedding layers, parameters), which however, is limited. One possible solution could be conducting a deeper analysis with open-source LLMs (e.g., LLaMA). Meanwhile, to figure out the mechanism behind ICL, conducting analysis on one single task (RE) may be insufficient since other NLP tasks also face the challenges of ICL. Thus, we find that another aspect could be to extend our proposed method to solve shortcomings of ICL in other NLP tasks, which can further improve our understanding of ICL. We will discuss the above issues in our future work section.

---

### Meta-Review · Area_Chair_NAvH · 2023-09-27

**Recommendation:** 5

**Metareview:**

The paper calls out the limitations of LLMs with in-context learning in performing the task of relation extraction, that they still lag behind supervised baselines such as fine-tuned BERT. It underlines a very valid concern regarding LLMs for this important problem. It outlines the limitations of these models and proposes an approach to mitigate them. The paper is very well-written and easy to follow. It provides a comprehensive survey of the related works in this space on four-widely used RE datasets. The community can benefit from this work for furthering the research in relation extraction using LLMs. The author responses are succinct and clarify most of the concerns of the reviews.

---

### Decision · Program_Chairs · 2023-10-07

**Decision:**

Accept-Main

**Comment:**

The paper calls out the limitations of LLMs with in-context learning in performing the task of relation extraction, that they still lag behind supervised baselines such as fine-tuned BERT. It underlines a very valid concern regarding LLMs for this important problem. It outlines the limitations of these models and proposes an approach to mitigate them. The paper is very well-written and easy to follow. It provides a comprehensive survey of the related works in this space on four-widely used RE datasets. The community can benefit from this work for furthering the research in relation extraction using LLMs. The author responses are succinct and clarify most of the concerns of the reviews.